# Teaching fish new tricks: Repeated exposure to a velocity barrier improves passage performance

Rachel MB Crawford[1,2]*, Eleanor M. Gee[2,¤a], Brendan J. Hicks[1,¤b],
Katherine A. Corn[3], Paul A. Franklin[2]

1 School of Science, Environmental Research Institute, The University of Waikato, Hamilton, New Zealand, 2 National Institute of Water and Atmospheric Research, Hamilton, New Zealand, 3 School of Biological Sciences, Washington State University, Pullman, Washington, United States of America

¤a Current address: Waikato Regional Council, Hamilton, New Zealand.
¤b Current address: Morphum Environmental Ltd, Hamilton, New Zealand.
* rachelmbcrawford@gmail.com

## Abstract

Instream structures like culverts and dams can impede upstream fish migration, acting as environmental filters that only allow onward migration of individuals that can successfully pass them. Cognition and learning ability may be an important factor in determining if a fish can successfully traverse such structures. This study investigated the effect of repeated exposure on passage performance of juvenile *Galaxias maculatus* through an experimental raceway. Over five consecutive days, individual fish were subjected to the same high-speed (0.45–0.5 m s$^{-1}$) conditions within the raceway, and performance on each day was recorded. The proportion of fish successfully passing the barrier increased significantly from 40% on Day 1–63% on Day 5. Time-to-event analysis further revealed that by Day 5, fish successfully passed the barrier at a significantly faster rate compared to Day 1. However, repeated exposure did not significantly improve approach or entry rates into the raceway. Fish length influenced approach rates, but not entry or passage rates. These findings suggest that cognition and spatial memory play a role in improving passage performance through velocity barriers, but other factors such as attraction flows may also play an important role in successful passage outcomes.

## Introduction

Instream structures such as culverts, dams, and weirs are known to act as "environmental filters" [1], selecting only the individuals that can swim past. As a consequence, potentially drastic effects on fish populations above and below the barrier can occur, including declines and extirpation [1–5]. These pressures can favour specific phenotypic traits, such as morphology, sex, physiology, and swimming performance, that may enhance a fish's ability to successfully pass barriers [5–8]. These effects are magnified by the separation of populations by such barriers,

**Data availability statement:** All data are currently available via the Zenodo digital repository, doi: https://doi.org/10.5281/zenodo.14194951.

**Funding:** New Zealand Ministry for Business Innovation and Employments [CO1X15] The University of Waikato Doctoral Scholarship. The funders had no role in study design, data collection and analysis, decision to publish, or preparation of the manuscript.

**Competing interests:** The authors have declared that no competing interests exist.

which has the potential to create phenotypically distinct populations above barriers compared to the populations below [9]. Recent research has suggested that instream structures create selective pressures not just on physical traits, but on behavioural traits as well. For instance, eels with more exploratory personalities have greater success traversing fishways than less exploratory eels, suggesting that behavioural filtering at fish passage barriers is a migratory impediment for cautious individuals [10].

Cognition, which is the process by which animals acquire, store, process, learn, and act on information, could also play a role in a fish's ability to traverse a barrier [11–16]. Cognition is required for spatial navigation, a crucial task for fish migrating through riverine habitat or in-stream structures [15]. Learning is a type of cognition in fishes that has been observed across various contexts, including foraging routes and behaviours, social behaviour, avoidance and recognition processes, spatial and landmark orientations, and migration [17–23]. Learning allows more flexibility in behaviour in a changing environment [20], and may be cognitively preprogramed and adapted to different environmental factors [24–27].

The ability of fishes to learn their spatial environment and monitor it is essential to their daily survival. Many fishes live in complex freshwater habitats, where water characteristics (pools and riffles), changes in substrate, woody debris, and vegetation make spatial cognition an important adaptation for fish [15]. When fish encounter a stimulus outside of their previous experience, they may use Hebbian learning to associate their behaviour and the environment [27]. Fish must continuously move through their environment and monitor their location with respect to external reference points, suggesting that they have significant spatial learning capacity [26]. Likewise, the ability of fish to detect environmental changes [28] and display organized exploration patterns when introduced to new environments [29], suggests the presence of spatial memory, which aids in their ability to adapt and thrive in their surroundings. Fishes are known to use this memory to both solve and remember spatial problems; for example, multiple exposures of guppies to a maze decreased the time and errors required to navigate it [15,30]. These observed effects of complex maze learning may also be due in part to classical conditioning. In experimental systems, fish received food and social rewards upon successful navigation of the maze, and in natural environments fish traverse complex landscapes to access food and conspecifics [15,27,31,32].

When encountering anthropogenic changes in the environment, it is hard to predict how fish might respond cognitively, and their ability to learn and overcome these challenges varies by situation, species, or individual [16,27,33]. Anthropogenic instream structures may require particularly complex spatial learning that combines response learning (going left or going right) and place learning (memorization of mental representation and landmarks, i.e., create a cognitive map) [34]. However, fishes appear to have some ability to learn and remember anthropogenic structures, which can mitigate the detrimental effects of anthropogenic stream structures and even save populations when human induced changes to migratory routes cause population decline [27]. Fish captured upstream of a fishway, having previously traversed it, are

more efficient at later traversing the fishway compared to downstream-captured fish that are naïve to the fishway [35–37]. Individual learning may, therefore, provide a crucial pathway for fishes to survive in a changing environment.

The role of learning as a response to anthropogenic structures is mainly understood in marine fisheries, classic model species, and migration of charismatic, large-bodied, freshwater fishes like salmonids. However, there remains little insight into obligately migratory amphidromous or catadromous species [37,38]. The capacity for learning to improve outcomes for fishes whose migration routes have been disturbed by artificial barriers suggests that it is important to understand the interaction between fish cognition and the navigation of structures in streams.

Our research aims to determine how repeated exposure influences an individual's ability to traverse a semi-permeable barrier such as a culvert with high water speed. In particular, we aim to improve understanding of the implications of multiple barriers on the migratory success of fishes. We studied the ability of learning to improve passage through a barrier in an obligate, amphidromous fish, *Galaxias maculatus,* that migrates upstream as a juvenile, and is widely distributed in the temperate Southern Hemisphere.

## Methods

*Galaxias maculatus* are typically amphidromous and migrate upstream from the sea as juveniles, so we used migratory juveniles for this work. The collection of fish for this research was conducted under the National Institute of Water and Atmospheric Research (NIWA) MPI special permit SP666−4. All experimental trials were conducted with approval from the NIWA Animal Ethics Committee (AEC204), adhering to the guidelines outlined in section 83 of the New Zealand Animal Welfare Act 1999.

### Fish collection and holding

Juvenile *Galaxias maculatus* were collected using whitebait nets and gee minnow traps from the Waikato River, New Zealand (−37° 18' 8.7156"S, 174° 50' 7.4688"E). Juvenile fish were captured at the beginning of their upstream migration to control for migratory urge across individuals. The fish were transported to the NIWA Hamilton fish laboratory in coolers containing aerated water sourced from their collection site. They remained in the coolers until the water temperature matched that of the laboratory tank water, after which they were transferred to 60 L quarantine tanks containing 6 ppt salinity to prevent disease. After one week, fish were transferred to 60 L freshwater tanks equipped with a recirculating water system. These tanks were located in a temperature-controlled room with a 12-hour light-dark cycle. The water temperature was maintained at 18°C±0.5°C. Fish were fed bloodworms every other day, with a 24-hour fasting period before experimentation to ensure they were in a postabsorptive state. Regular monitoring of ammonia and pH concentrations was conducted, and water changes were performed when ammonia levels reached or exceeded 0.25 mg L$^{-1}$.

### Experimental setup

Experiments were conducted using an indoor recirculating flume. The flume, constructed from acrylic, had dimensions of 7.5 m in length, 0.5 m in width, and 0.5 m in depth, capable of holding up to 1500 L of water in its working section. Beneath the flume ran a return pipe measuring 0.4 m in diameter. To ensure smooth flow, a flow straightener made of stacked plastic polyvinyl chloride (PVC) tubes, each 0.02 m in diameter and 0.3 m long, was positioned at the upstream end of the flume. Water speed was regulated by an impeller driven by a variable-speed AC motor installed in the descending section of the return pipe. Spot water temperature measurements were taken using a handheld thermometer at the beginning and conclusion of each trial. Across all trials, flume water temperatures ranged from 17.9°C to 22.2°C, with a mean of 21.1°C (standard deviation: 0.96°C). While other environmental variables were not monitored, such as dissolved oxygen and pH levels, we expected the variability in these variables to be relatively minimal. The flume was open to the air and had turbulent flow (Reynolds number=220, 588). Any differences in dissolved oxygen would be minimal and

strongly correlated with temperature. For pH, the fish were very small relative to the volume of the experimental setup, and no other factors were present that would influence pH, as the system used raw, dechlorinated, water.

A raceway was built in the flume to constrict the flow, creating a water speed challenge for the fish. The design ensured that the maximum water speed in the constricted section ranged from 0.45 and 0.50 m s⁻¹. The water speed was chosen based on Crawford et al. [39], which demonstrated that 90% of *G. maculatus* individuals had critical swimming speeds of 0.45 m s⁻¹ or less. Ideally, fish would traverse anthropogenic barriers at or below their critical swimming speed to avoid excessive physiological stress. However, in natural settings, fish often resort to burst swimming when encountering high-velocity barriers, which can only be maintained for a matter of seconds and is fuelled entirely anaerobically [40]. By choosing a water speed that exceeds the critical swimming speed of 90% of individuals, we ensured that most fish were required to use burst swimming to pass through the artificial raceway. The test section spanned from 0.0 m to 5.9 m within the working section of the flume. Constructed from PVC (see Fig 1), the raceway comprised two sets of angled wingwalls, each 1.2 m in length, positioned at a 40° angle relative to the flume wall, both upstream and downstream. The angle of 40° was chosen to provide a relatively smooth transition into and out of the raceway, allowing for a smoother acceleration of flow without introducing much turbulence.

The body of the raceway measured 2.9 m in length and 0.2 m in width, with a consistent water depth of 0.2 m maintained throughout all trials. Mesh screens were installed upstream and downstream of the raceway to contain the fish within the experimental area. The base of the raceway was marked with tape at 0.1 m intervals, starting from 0.0 at the entrance to the downstream wingwalls. Additional markings denoting the approach, entry, and passage thresholds within the raceway (0.6, 1.8, and 4.7 m, respectively) were positioned from the end of the downstream working section within the flume (Fig 1).

For analysis of passage, the raceway was delineated at three points, which the entire body of the fish had to pass to count as success (Fig 1):

• approach threshold – marking the threshold into the approach of the raceway (Fig 1, 1A);

• entry threshold – marking the threshold into the entry of the raceway (Fig 1, 1B);

• passage threshold – marking the threshold out of the upstream end of the raceway (Fig 1, 1C).

To assess the variation in water speed within the raceway, three-point vertical profiles of water speed were measured at 0.5 m intervals along the centre of the channel, starting 0.2 m from the downstream end of the raceway, along the axis of

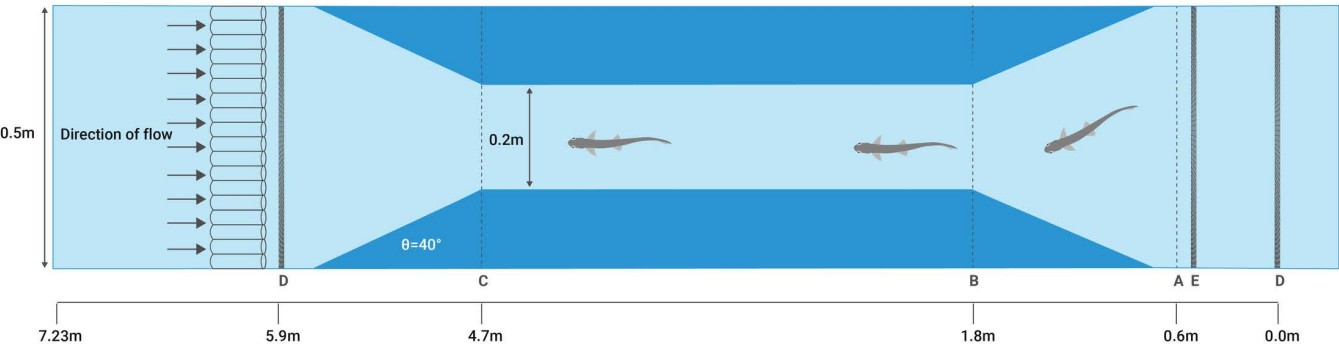

**Fig 1. Aerial view experimental setup, focusing on the working section containing the raceway.** (A) Line marking the "approach" to the raceway, (B) line marking the "entry" to the raceway, (C) line marking successful "passage" of the fishway, (D) mesh screens (visible as dark bars) to confine fish within the flume, (E) removeable mesh screen, containing fish in the "starting section" for acclimation before the start of the trial. No change in bed elevation throughout the entirety of the flume. Water depth across the whole flume was 0.2 m throughout all trials.

the flume in the direction of flow. These profiles were generated using a Sontek Flow Tracker 2 Acoustic Doppler Velocimeter (ADV). Measurements were taken at three heights: 20%, 60%, and 80% of the water column above the raceway floor. Water speed data, measured in m s$^{-1}$, were collected for 40 s at each height, with a sampling frequency of 10.0 MHz. Following data collection, the FlowTracker2 desktop software was used to average the water speed across each depth at each station and visualize the water speeds experienced by the fish (Fig 2).

Five Swan security cameras were positioned 1.1 m above the flume floor, evenly spaced along the length of the working section with a slight overlap between camera frames. To track fish movement accurately, video recording from each camera was synchronized throughout the trial duration. All recordings were captured in colour, with all cameras set to a consistent frame rate of 30 frames per second.

## Experimental protocol

A total of 40 trials were conducted, each trial consisted of an individual fish and no fish were reused: mean length of 51.7 mm ± 8.97; mean weight 0.48 g ± 0.32; mean condition factor 0.26 ± 0.09. Each trial was conducted over five consecutive days, where fish were exposed to the same experimental conditions once per day over this five-day period. Prior to their introduction into the flume, the body length and weight of each fish was recorded. The condition factor of each fish was calculated using Fulton's condition factor formula [41]. Fish weight and condition factor were not included in any of the statistical models, after calculations showed a high correlation with length (0.85 and 0.76 respectively). At the start of the trial, fish were placed in the starting section, with a mesh screen in place to prevent them from entering the raceway (Fig 1, 1D & 1E). The fish were acclimated in the starting section, where the water speed was maintained at 0.0 m s$^{-1}$, for a duration of 30 minutes to mitigate the potential effects of handling stress on the fish.

The cameras recorded fish movement for the duration of each 30-minute trial. The impeller motor was set to 31 Hz, ensuring water speed was between 0.45 and 0.5 m s$^{-1}$ within the working section of the raceway. Once the speed in the flume was constant through the working section, the downstream mesh screen was removed (Fig 1, 1E). Throughout the trial, the times at which fish crossed the approach, entry, and passage threshold were recorded, based on time-stamped

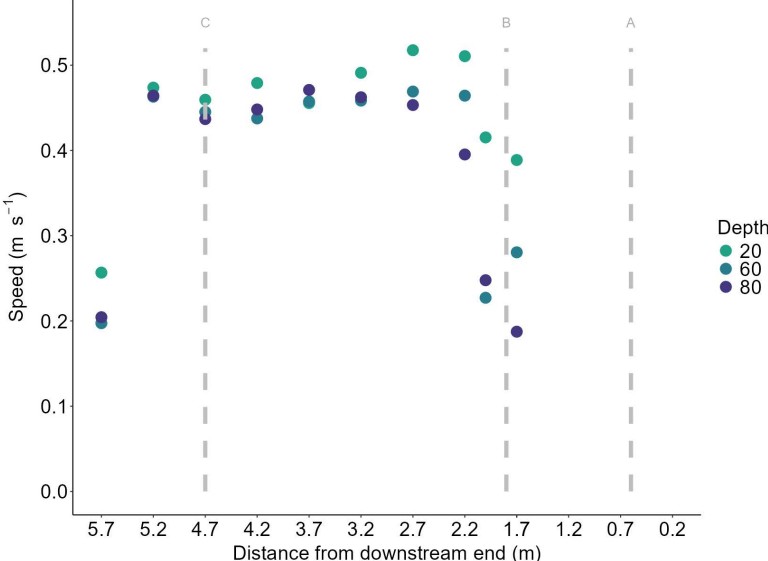

**Fig 2. ADV measurements throughout the raceway taken at 20%, 60%, and 80% depth of the water column.** Grey, dashed lines represent the fish, approach, and entry to the raceway from left to right.

video footage to the nearest second. To avoid duplicating data, only the first attempt for each fish was recorded, although fish generally did not make multiple attempts at each threshold. Trial duration was set to a maximum of 30 minutes based on preliminary observations indicating that fish that had not successfully passed through the raceway within this period were unlikely to do so even after several additional hours. This duration also allowed for the practical scheduling of multiple trials per day. After each trial, fish were returned to the holding tank and allowed to recover until the subsequent trial, conducted 24 hours later.

## Analysis

All analyses were carried out in R version 4.2.0 [42].

## Binary success model

Success was modelled with binary logistic regression, where a fish was given a 1 for successfully crossing a threshold or a 0 for failing to cross a threshold. To model the effect that repeat exposure to a barrier had on the success of fish passing each threshold (approach success, entry success, passage success), a binomial Generalized Linear Mixed Effects Model (GLMM) model was used. Success (0 or 1) was the response variable, day (representing number of exposures) and fish length were fixed predictor variables, and fish number was included as a random predictor variable. For these models, a logit link function was selected without adjusting for overdispersion. A separate model was applied to each threshold of the raceway. Analysis was carried out using the glmmTMB package in R [43]. Model fit was compared using corrected Akaike Information Criterion (AICc), starting with a fully saturated model. The selected model for passage success had a ΔAIC more than two units lower than the other candidate models. The selected models for approach and entry success had a ΔAIC less than two units lower than the other candidate models, therefore the most parsimonious model with no interaction was selected (Table 1) [44].

## Time-to-event analysis

We used time-to-event analysis, also known as survival analysis, to assess the rate of fish attempting the various stages within the raceway (approach rate, entry rate, passage rate) [45,46]. Time-to-event analysis enables the inclusion of censored data, allowing us to account for fish that did not successfully pass each threshold of the raceway by the end of the 30 min trial. Time-to-event analysis calculates the likelihood that a fish will cross the threshold within the raceway at a particular time, given that it hasn't yet crossed that threshold (i.e. the event rate). We refer to the event rate at the approach threshold, entry threshold, and passage threshold as the "approach rate," "entry rate," and "passage rate," respectively.

**Table 1. Model selection results for the assessment of success at each threshold. LogLik is the log likelihood, K is the number of parameters +1, corrected Akaike Information Criteria (AICc).**

| Model | LogLik | k | AICc | ΔAICc |
|---|---|---|---|---|
| **Approach success** | | | | |
| approach success ~ Day * Length + (1|Fish) | −93.32 | 5 | 195.64 | 0 |
| approach success ~ Day + Length + (1|Fish) | −93.72 | 4 | 195.64 | 0.00 |
| **Entry success** | | | | |
| entry success ~ Day * Length + (1|Fish) | −130.56 | 5 | 271.42 | 0 |
| entry success ~ Day + Length + (1|Fish) | −131.24 | 4 | 270.70 | 0.72 |
| **Passage success** | | | | |
| passage success ~ Day * Length + (1|Fish) | −131.51 | 5 | 271.36 | 0 |
| passage success ~ Day + Length + (1|Fish) | −130.52 | 4 | 268.74 | 2.62 |

The Cox regression model was used to model instantaneous event rate at each threshold, which was right censored and modelled as a function of time as shown in the following equation, and explained by [46]:

$$\lambda(t) = \lambda_0(t)e^{X\beta + Zb}$$

where $\lambda(t)$ is the baseline hazard function (i.e. event rate) modelled as a function of time (t), X represents the matrix of fixed effects, Z represents the matrix of random effects, $\beta$ and $b$ represent the fixed- and random-effect coefficients, respectively. Day and fish length were the fixed effects, with fish number included as a frailty term to account for random effects. The chosen model was tested using scaled Schoenfeld residuals and met Cox proportional hazard model assumptions. Based on this equation, hazard ratios (HR) were calculated for each treatment, comparing the event rates between days over time (representing the number of exposures). A hazard ratio greater than 1 indicates that the treatment (repeat exposure) has a higher event rate compared to the reference; a hazard ratio less than 1 indicates the treatment (repeat exposure) has a lower event rate compared to the reference (day 1); a hazard ratio equal to 1 indicates no difference in event rate between treatments.

These models were implemented using the Coxme package in R version 4.2.0 [42,47]. The Likelihood Ratio Test, Wald Test, and Score (Logrank) Test were used to identify model fit. We developed three distinct models to answer specific questions:

1. The first model addresses the approach rate: the likelihood of a fish to cross the approach threshold per unit of time.

2. The second model addresses the entry rate: the likelihood of a fish to cross the entry threshold per unit of time.

3. The third model addresses the passage rate: the likelihood of a fish to cross the passage threshold per unit of time.

We then used the survfit function form the Coxme package to calculate the proportion of fish that are predicted to cross each threshold at a given time for each event rate model. As time progresses, the predicted proportion of fish crossing the threshold decreases because the number of available fish still to cross the threshold diminishes over time.

## Results

Modelling binary success Seventy-five percent of fish successfully approached the raceway on Day 1, compared to 78% on Day 5 (Table 2, Fig 3). There was an increase in fish successfully approaching the raceway on Day 2 (90%), however this increase was not statistically significant and did not hold steady throughout the rest of the days. There was no statistically significant effect of repeated exposure on approach success (p=0.998, Table 3). Fish length was found to be a statistically significant factor influencing approach success, with larger fish more likely to approach than smaller fish (p=0.01, Table 3).

On Day 1, 58% of fish successfully entered the raceway, compared to 63% on Day 5 (Table 2, Fig 3). There was no statistically significant effect of repeated exposure on entry success (p=0.38, Table 3), and no statistically significant effect of fish length on entry success (p=0.72, Table 3).

**Table 2. Percent success across all five days.**

| Day | Percent Success | | |
|---|---|---|---|
| | Approach | Entry | Passage |
| 1 | 0.75 | 0.58 | 0.4 |
| 2 | 0.9 | 0.58 | 0.45 |
| 3 | 0.78 | 0.65 | 0.58 |
| 4 | 0.83 | 0.68 | 0.7 |
| 5 | 0.78 | 0.63 | 0.63 |

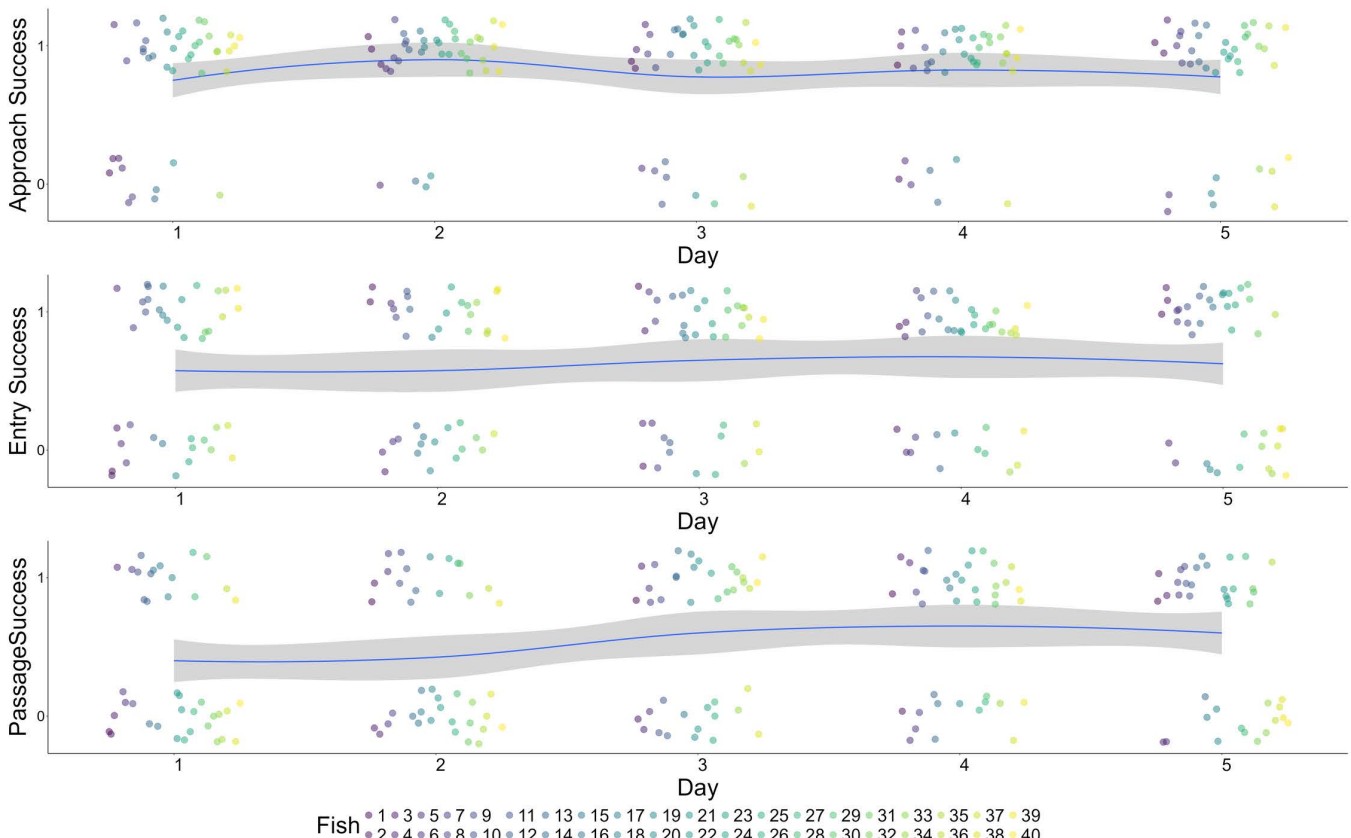

**Fig 3. Binomial logistic regression of approach success, entry success, and passage success across five days.** Each fish is represented by a point.

**Table 3. Model output for binary logistic regression models for all three raceway thresholds.**

| Parameter | Coefficient estimates | Standard error | Z value | P value |
|---|---|---|---|---|
| **Approach success** | | | | |
| Day | 0 | 0.13 | −0.02 | 0.98 |
| Fish length | −0.08 | 0.03 | −2.59 | 0.01 |
| **Entry success** | | | | |
| Day | 0.1 | 0.11 | 0.88 | 0.38 |
| Fish length | −0.01 | 0.02 | −0.35 | 0.72 |
| **Passage success** | | | | |
| Day | 0.25 | 0.11 | 2.38 | 0.017 |
| Fish length | 0.05 | 0.02 | 2.38 | 0.017 |

On Day 1, 40% of fish successfully passed the raceway compared to 63% on Day 5 (Table 2, Fig 3). There was a statistically significant effect of repeated exposure on passage success (p = 0.017, Table 3), indicating that subsequent exposures to the barrier positively affected passage success. There was also a statistically significant effect of fish length

on passage success, with larger fish more likely to pass than smaller fish (p = 0.017, Table 3). Further examination of the data revealed that there was a higher percentage of passage success on Day 3 and Day 4 compared to Day 1 (Table 2). However, this increase appeared to asymptote and even decrease after Day 4, with no significant difference between Day 4 and Day 5.

### Model for event rate – Time to event analysis

An average of 8 out of 40 fish failed to approach the raceway each day (range: 4–10 fish) but were still included in the analysis and censored at the maximum trial time (30 min). There was no statistically significant effect of repeat exposure (day) on approach rate (p = 0.99, HR = 1.0; Table 4, Fig 4).

Day 1 fish had a median approach time of 325 s and Day 5 fish had a median time of 312 s (Table 5, Fig 3). There was a statistically significant effect of fish length on approach rates (p = 0.002), but the hazard ratio indicated neither a positive nor negative effect of fish length on outcome (HR = 0.97; Table 4).

An average of 15 out of 40 fish failed to enter the raceway each day (range: 13–17 fish) but were still included in the analysis and censored at the maximum trial time. There was no statistically significant effect of repeat exposure (day) on entry rate (p = 0.46, HR = 1.03; Table 4, Fig 4). Day 1 fish had a median entry time of 548 s and Day 5 fish had a median time of 207 s (Table 5, Fig 3). There was no statistically significant effect of fish length on entry rates (p = 0.97, HR = 1.14; Table 4).

An average of 18 out of 40 fish failed to pass the raceway each day (range: 12–24 fish) but were still included in the analysis and censored at the maximum trial time. There was a statistically significant effect of repeat exposure (day) on passage rate (p = 0.021, HR = 1.17; Table 4, Fig 4). Median passage time was greater than the trial length on Day 1, and Day 5 fish had a median time of 42 s (Table 5, Fig 3). There was no statistically significant effect of fish length on passage rates at the p < 0.05 threshold (p = 0.11, HR = 1.02; Table 4). Passage success rates asymptoted by Day 5.

## Discussion

Human modifications of freshwater systems have increased the challenges faced by the millions of fishes migrating each year. In particular, fish migrating upstream to their spawning or rearing grounds may face several instream structures along a single river network that either completely or partially impede their upstream migration [48]. As migrants encounter several of these barriers on their way upstream, it is important to understand how previous exposure to a barrier may influence subsequent exposures. In this study, we repeatedly exposed fish to the same barrier as a proxy for the

**Table 4. Estimation of parameters for the selected event rate model for each raceway threshold.**

| Parameter | Coefficient ± SE | Hazard Ratio | Chi² | Degrees of Freedom | P value |
|---|---|---|---|---|---|
| **Approach rate** | | | | | |
| Day | −0.0004 ± 0.06 | 1.00 | 0 | 1 | 0.99 |
| Fish length | −0.03 ± 0.007 | 0.97 | 9.9 | 1 | 0.002 |
| Frailty (Fish number) | – | – | 21.04 | 13.52 | 0.09 |
| **Entry rate** | | | | | |
| Day | 0.047 ± 0.06 | 1.03 | 0.54 | 1 | 0.46 |
| Fish length | −0.0004 ± 0.01 | 1.14 | 0.04 | 1 | 0.97 |
| Frailty (Fish number) | – | – | 11.76 | 9.03 | 0.23 |
| **Passage rate** | | | | | |
| Day | 0.15 ± 0.07 | 1.17 | 5.35 | 1 | 0.021 |
| Fish length | 0.02 ± 0.01 | 1.02 | 2.59 | 1 | 0.11 |
| Frailty (Fish number) | – | – | 0 | 0 | 0.92 |

cumulative effect of encountering multiple barriers during natural migration. We exposed local, obligately migratory fishes to a water velocity barrier and found that individuals were more successful at passing through the barrier when they had been exposed to it multiple times. Specifically, a higher proportion of individuals successfully pass the barrier on Days 3 and 4 when compared to Day 1, and overall passage rates on Days 3, 4, and 5 were higher when compared to Day 1. The higher passage rates were especially evident when looking at the median time threshold, where on Day 1, 50% of fish

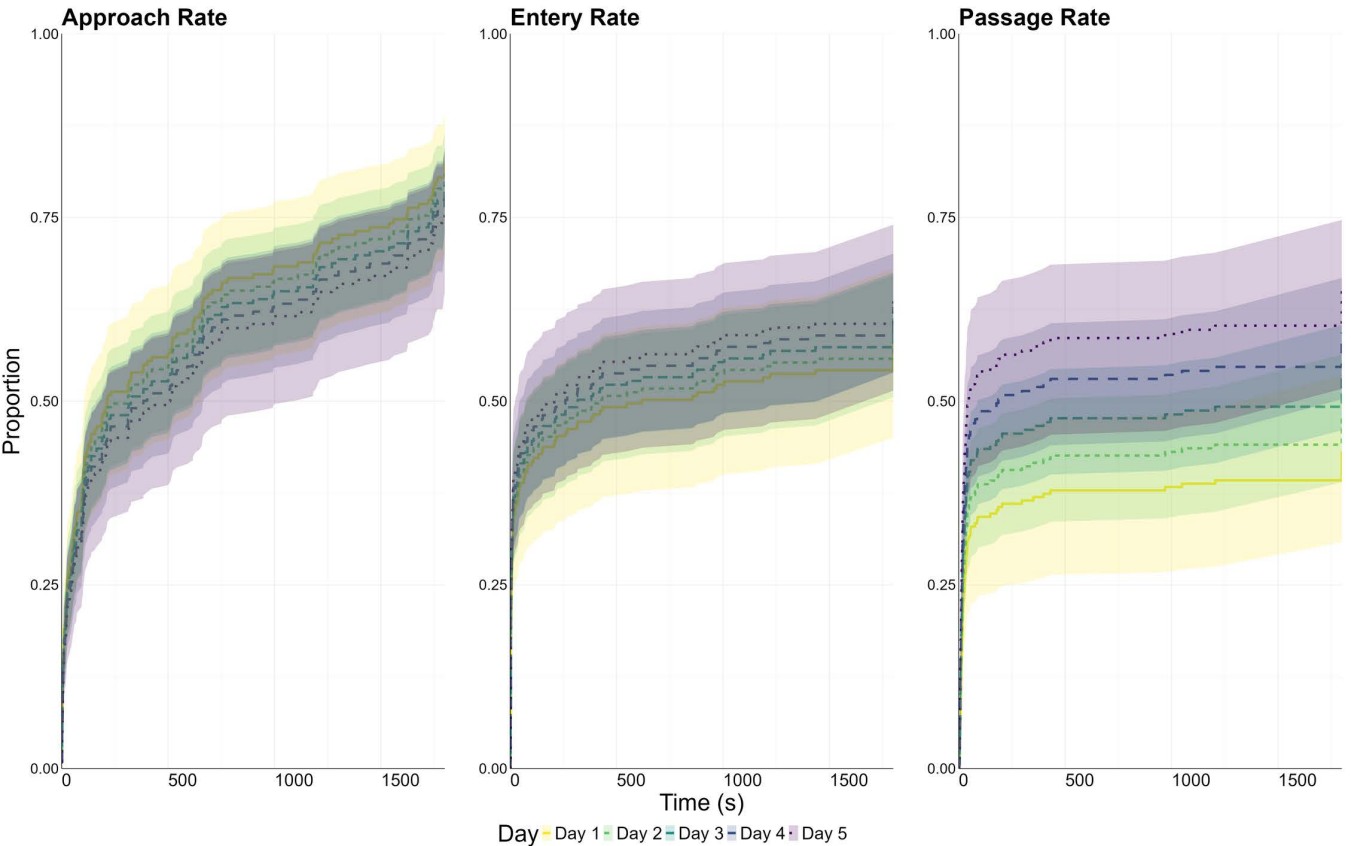

**Fig 4. Event rate models for each raceway threshold, representing the proportion of fish crossing each threshold as a function of time.** Zero on the y-axis represents that 0% of the population have crossed the threshold at that time (t), and one represents that 100% of the fish have crossed the threshold at time (t). Data are stratified by exposure day (Day 1-5). When the lines diverge, this indicates a hazard ratio that differs from 1. Where the lines overlap, it suggests a hazard ratio of 1, indicating no significant difference between exposure days. Shaded regions in the corresponding colours represent the 95% confidence intervals.

**Table 5. Median time to threshold values for the selected event rate model across all days. Null values in the table represent days where less than 50% of fish were successful. Total trial time was 1800 s.**

| Exposure Day | Median time to threshold (s) | | |
| --- | --- | --- | --- |
| | Approach | Entry | Passage |
| Day 1 | 325 | 433 | – |
| Day 2 | 325 | 390 | – |
| Day 3 | 315 | 315 | 1205 |
| Day 4 | 315 | 217 | 177 |
| Day 5 | 312 | 207 | 42s |

failed to pass the barrier during the 30 min trial, and 50% of fish successfully passed the barrier within just 42 s on Day 5. This suggests that not only do more fish successfully pass the barrier after repeated exposure, but they also pass the barrier more quickly. Our study looked at relatively short passage periods of 30 min compared to other research that looks at passage time over longer timescales in fishways. However, passage time through a barrier has been found to be an important factor influencing fish survival and migration success. Prolonged delays in passing barriers have been linked to higher rates of predation, increased energy demand, decreased migration speed, and decreased motivation [49–54].

The significant increase in passage success and passage rates observed is consistent with a pattern in which fish learn the raceway through spatial learning or landmark cues. Cognition in fishes is best understood as a combination of spatial learning and problem-solving, as it relates to visual cues and well-structured habitats [25–27,24]. Fishes are routinely observed learning and remembering mazes in laboratory settings, where the structural complexity of the experimental setup simulates the complex aquatic habitat that they must migrate through [15,30]. These indicators of spatial learning and memory in cognition, are supported by observations of faster navigation times upon repeated exposures to the maze [30,33]. Consistent with maze-based studies, our results suggest that repeated exposure to an instream barrier may improve passage outcomes in the wild. It is important to note that the use of an experimental raceway may not fully replicate the complexities of real-world structures and flow conditions, especially at the scale used in this study. However, the controlled environment of the laboratory flume offers several advantages, including the ability to control for confounding factors. This allows us to more accurately isolate learning effects from other influences, such as variation in group size, as well as ensuring consistent flow conditions and water speeds across all trials. Additionally, as only one species was tested, care should be taken not to extrapolate potential cognitive abilities to other fish species.

Our study indicates that prior exposure to a barrier can enhance passage success. As such, where fish encounter multiple barriers along a migration route there is potential that previous experience may improve the likelihood of passage at successive barriers. However, in the real world, the characteristics of each barrier are unique. It remains unclear how the observed improvement in passage success following repeat exposure to the same barrier would translate to scenarios involving different barriers As we did not observe a 100% improvement in passage success across all individual fish, nor a significant improvement in approach or entry with repeated exposure, our results suggest that fish passage solutions require a more nuanced approach to design in order to enhance their effectiveness. There is no "one size fits all" approach as fish passage performance is influenced by a range of factors [39,55,56]. These include variation in swimming ability both between and within species, environmental conditions such as changes in water temperature, and behavioural factors like group swimming [57,58], and potential variation in fish cognition as highlighted in this study. Accounting for these sources of variation is important for ensuring successful fish passage [39,55,56]. We observed a 50% increase in fish passage success over the five-day trial, suggesting memory and cognition played a role in success. However, the fact that passage success never reached 100% indicates other factors influenced the ability of fish to traverse the barrier as well. Physical and behavioural traits, such as fish size and health, are known to affect swimming performance and may have also affected passage success [5,40,59,60]. Although we did not observe a significant effect of fish length on passage rates at the $p = 0.05$ threshold, we observed that fish length significantly influenced both approach rates and overall success of approach and passage. These results support the idea that physical characteristics may play an important role in passage success.

The positive effects of subsequent exposure on success were not mirrored in the approach and entry to the raceway. The overall rates of fish approaching the fishway remained consistent over time, and the median time to the approach threshold did not vary by more than 13 seconds throughout the five-day trial. Although we did not directly assess fish boldness or aversion, the observed pattern suggests that repeated exposure to the fishway may intensify the effects of shyness and aversion in individuals less inclined to approach or enter the raceway Previous studies have found a correlation between personality traits and the likelihood of approaching a challenge, with bolder individuals being more likely to succeed [10,61–63]. Bold individuals tend to commit more quickly to actions and show less variability in their movements [64]. The lack of a clear effect

at the approach threshold suggests that spatial cognition alone did not drive behaviour at this stage; rather, it may be related to the individual's boldness. Consequently, even with repeated exposure to the same barrier, shy fish may be less likely to explore and may never cross the approach threshold on subsequent trials.[10]. The lack of improvement in approach and entrance success over time highlights a key issue raised in [35] meta-analysis on fishway efficiency. They demonstrated that passage efficiency was higher than attraction efficiency across species, migratory stages, and ecological niches. [34] also found that efficiency was especially low for small-bodied fish. This has been corroborated by [65], whose meta-analysis found that fishway type significantly influences the attraction efficiency of non-salmonids and that these fish are more sensitive to hydrological variations, such as attraction flow, backflow, hydraulic jump, and large-scale vortices. This suggests that for most species, locating and entering a fishway may pose a greater challenge than successfully traversing it [21]. Efforts to attract small-bodied, non-salmonid fish for successful passage require greater attention, especially given the global use of large-scale fishways in areas where non-salmonid species are present.

## Conclusion

In this study, we highlight the complex dynamics of fish passage through anthropogenic barriers in freshwater systems. We found that repeated exposure to a barrier improves passage rates and time to success, suggesting that fish possess a capacity for learning to traverse such barriers, potentially through spatial learning or landmark cues. However, while this positive effect of repeated exposure on passage success was statistically significant, it was marginal, indicating that other factors, including non-learned behaviours or physiological traits, such as fish size, may also play a role in passage success. Our findings highlight the importance of considering both learned and innate factors when assessing fish passage through barriers. We also highlight the challenges of attracting fish to fishway structures, particularly for small-bodied species. Efforts to improve attraction efficiency for small-bodied fish are essential for ensuring successful passage through anthropogenic barriers.

## Acknowledgments

Thank you to Ted Castro-Santos, Shad Mahlum, and Elizabeth Graham for taking the time to help with the statistical analyses, particularly the time-to-event analysis. Thanks to Emily White for helping with fish collection. Thanks to Emily White and Rochelle Petrie for helping with the fish handling. Thanks to Peter Williams and Gordon Tieman for helping with experimental setup.

## Author contributions

**Conceptualization:** Rachel MB Crawford, Eleanor M Gee, Brendan J Hicks, Paul A Franklin.

**Data curation:** Rachel MB Crawford.

**Formal analysis:** Rachel MB Crawford.

**Funding acquisition:** Paul A Franklin.

**Investigation:** Rachel MB Crawford.

**Methodology:** Rachel MB Crawford, Eleanor M Gee, Brendan J Hicks, Paul A Franklin.

**Project administration:** Rachel MB Crawford.

**Supervision:** Eleanor M Gee, Brendan J Hicks, Paul A Franklin.

**Visualization:** Rachel MB Crawford.

**Writing – original draft:** Rachel MB Crawford.

**Writing – review & editing:** Eleanor M Gee, Brendan J Hicks, Katherine A Corn, Paul A Franklin.

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
