## [Decision Letter · Decision Letter 0]

24 Mar 2025

Dear Dr. Crawford,

Thank you for submitting your manuscript to PLOS ONE. After careful consideration, we feel that it has merit but does not fully meet PLOS ONE’s publication criteria as it currently stands. Therefore, we invite you to submit a revised version of the manuscript that addresses the points raised during the review process.

We look forward to receiving your revised manuscript.

Kind regards,

Mizanur Rahman, Ph.D.

Academic Editor

PLOS ONE

Journal Requirements:

 “New Zealand Ministry for Business Innovation and Employments [CO1X15]

The University of Waikato Doctoral Scholarship” 

Reviewers' comments:

Reviewer's Responses to Questions

**Comments to the Author**

1. Is the manuscript technically sound, and do the data support the conclusions?

Reviewer #1: Yes

Reviewer #2: Yes

Reviewer #3: Yes

Reviewer #4: Partly

2. Has the statistical analysis been performed appropriately and rigorously?

Reviewer #1: Yes

Reviewer #2: Yes

Reviewer #3: Yes

Reviewer #4: Yes

3. Have the authors made all data underlying the findings in their manuscript fully available?

Reviewer #1: Yes

Reviewer #2: Yes

Reviewer #3: No

Reviewer #4: Yes

4. Is the manuscript presented in an intelligible fashion and written in standard English?

Reviewer #1: Yes

Reviewer #2: Yes

Reviewer #3: Yes

Reviewer #4: Yes

Reviewer #1: Comments on the manuscript

Thank you for the opportunity to review this manuscript. The paper is well-written and structured, and it addresses an important topic: how repeated exposure to a high-velocity water barrier affects the ability of juvenile Galaxias maculatus to pass through it. The experiment involved subjecting individual fish to the same high-speed water conditions in a raceway over five consecutive days. They found that the proportion of fish successfully passing the barrier increased significantly from 40% on Day 1 to 63% on Day 5. Time-to-event analysis revealed that fish passed the barrier more quickly by Day 5 than Day 1 but did not significantly improve approach or entry rates into the raceway. The study suggests that cognition and spatial memory play a role in improving passage performance through velocity barriers, but other factors, such as attraction flows, are also important. However, there are a few points that could be strengthened:

1. A total of 40 fish were used in the experiment. Although this sample size may be adequate, a larger sample size could provide more robust results and increase the statistical power of the study.

2. The study focuses on one species (Galaxias maculatus), which may not fully represent cognitive responses in other fish species.

3. While the study discusses cognition and memory, it does not explicitly assess fish personality traits (e.g., boldness or exploratory behavior), which could influence passage success.

4. While water temperature was monitored, other environmental factors, such as dissolved oxygen and pH levels, were not explicitly addressed. Fluctuations in these factors could potentially influence fish behavior and passage performance.

5. The experimental raceway does not fully replicate natural stream conditions, which may affect generalizability.

6. The study does not assess whether repeated exposure induces stress or fatigue, which could influence fish behavior.

Therefore, based on the concerns outlined above, I recommend that this manuscript be accepted for publication after major revision.

Reviewer #2: The manuscript presents a well-designed and executed study with important implications for understanding fish passage through anthropogenic barriers. However, the ecological realism, exploration of behavioral mechanisms, and statistical reporting could be improved. Addressing these weaknesses would strengthen the manuscript and enhance its contribution to the field of freshwater ecology and conservation.

Reviewer #3: 1. The manuscript presents a technically sound scientific investigation, supported by a detailed and rigorous methodology. It includes all the essential elements of a well-structured study, such as the description of the required permits and approvals for fish collection, experimental trials, setup and experimental protocol, as well as statistical analysis.

The results are presented clearly and in alignment with the methodology, allowing the reader to assess whether the objectives were achieved. Additionally, the analyses are well-founded, providing a robust interpretation of the data while transparently acknowledging the study’s limitations. In particular, the article highlights the ability of fish to improve their passage success with repeated exposures but also recognizes that it remains uncertain whether this learning is transferable to new obstacles.

A valuable aspect of the manuscript is that, in addition to addressing its research question, it provides relevant findings for the management of fish passage structures. It highlights challenges in fish attraction and entry, as well as the potential effect of shyness in some individuals, suggesting that management strategies should focus not only on the physical structure of passageways but also on improving fish attraction.

Overall, this study significantly contributes to the understanding of fish passage efficiency and provides valuable information for both science and the management of aquatic ecosystems.

Editorial Recommendations

Some structural adjustments are necessary in the manuscript:

a) In the Modelling Binary Success section, when mentioning p-values, the text should correctly direct the reader to Table 2.

b) In line 278, "Table 2" should be changed to "Table 4."

c) It is recommended to reorganize the order of the paragraphs to maintain the logical sequence of the results, placing Table 3 before Table 4.

2. I consider that the statistical analysis in the manuscript has been conducted appropriately and rigorously. The authors applied a solid approach using generalized binomial generalized linear mixed models (GLMM) to model binary success in approach, entry, and successful passage, incorporating random effects to capture individual variability in fish. Additionally, they employed Cox regression to assess event rates (approach rate, entry rate, and passage rate), allowing for the interpretation of the effects of repeated exposure over time and generating Hazard Ratios (HR) to measure the instantaneous probability of success at each threshold of the experimental channel.

However, I found an inconsistency in the application of the Hosmer-Lemeshow goodness-of-fit test. In the methodology, it is stated that this test was used to evaluate model fit in the GLMM for all three events (approach, entry, and successful passage), but in the results, its use is only reported for the entry model. This raises questions about whether it was also applied to the other two events or if its reporting was omitted.

Furthermore, I consider that the Hosmer-Lemeshow test is not the most appropriate for evaluating the fit of GLMM models, as it does not account for the random effects characteristic of these models. I suggest using more suitable methods, such as residual inspection, AIC/BIC comparison, or the simulation of predicted versus observed values. If a new model fit method is adopted, I recommend applying it uniformly across all events and including the corresponding results in the results section to ensure consistency with what is described in the methodology.

Despite this observation, I believe that the analyses used in the study are appropriate for addressing the research question and allow for well-supported conclusions.

3. In the manuscript, the authors indicate that the data will be available through the Zenodo digital repository and provide the corresponding doi: 10.5281/zenodo.14194951. However, I was unable to access this DOI or find an active link.

I suggest that the authors verify the availability of the link and confirm that the data are correctly deposited in the repository to ensure open access and transparency in the publication.

4. The manuscript is written clearly, without ambiguities, and in a comprehensible level of English. During my review, I did not identify any typographical or grammatical errors that would affect the understanding of the content. However, since English is not my native language, my assessment of full compliance with linguistic standards is limited.

Reviewer #4: L105

Were the juveniles caught before they had completed their upstream migration to their habitat? Or was the capture location their habitat?

This may be considered one of the reasons why "fish failed to enter the raceway each day."

L117

Did they reuse the waterway described in (39), or did they use the features of (39)?

I would like you to add a little more explanation about the introduction of the waterway so that it can be understood without reading (39).

If PVC is an abbreviation for something, I would like an explanation for that.

L130

If the subject of the paper is "velocity barrier," I think there should be some experiments or discussion on "rush speed" as well as "critical swimming speed."

L134

It would be good if the design concept of the "40° angle" was described.

I would like to see the concept of the water flow not starting from a single point in the center, but evenly flowing from a 0.5m width or from multiple outlets described.

L137

It is suggested that the "starting pool" refers to the space between (D) and (E) on the right, but a more specific description would be needed.

If there was an elevation drawing, it would be easier to understand the following points:

- "with a consistent water depth of 0.2 m maintained throughout all trials."

- "Measurements were taken at three heights: 20%, 60%, and 80% of the water column above the raceway floor."

- The height at which the starting water flow occurs

Does "successful "passage" of the fishway" mean the passage of the entire fish body?

L165

Please state in the caption of the image that the water depth is 50 cm or give a specific water depth.

L180

I would like to know if "Fish weight and condition factor were not included in any of the statistical models due to high correlation with length (0.85 and 0.76 respectively)." is a general statement or a result calculated this time.

L182

Does "in the downstream pool" refer to the "starting pool"?

L212

Regarding "the end of the 30 min trial," I was unable to find an explanation for why the maximum trial time was set at 30 minutes.

Please describe how the juvenile fish were managed between experiments (23 hours?).

L220

It seems to me that this equation only applies to fish where the relationship between body length and weight is constant, such as "young Plecoglossus altivelis" or "young Oncorhynchus masou" but will this be considered?

L251

If there were a clear explanation somewhere that larger fish are easier to approach, this statement would be easier to understand, and it could also be used in discussions about small fish.

L260

Although Figure 3 appears before Table 2 in the text, why are they listed in the opposite order?

L264

Considering the development of the discussion in this study, if the "effect of day" could be rephrased somewhere with the intention of "effect of frequency," mutual understanding would advance.

L265

Considering this case in which the fish's body length has a significant effect, it would be helpful to clarify the relationship between the fish's body length and their charging speed.

L268

Does "this increase appeared to asymptote and even decrease after Day 4," indicate that while repeated attempts increase the probability of success, repeated attempts also cause some individuals to give up on the upstream migration?

L276

I would like to know which values in Table 2 are used and how the calculation results in "HR = 1.0."

I did not understand how Figure 4 should be interpreted to support this statement.

L279

Although Figure 4 appears before Table 3 in the text, why are they presented in the opposite order?

L314

I felt that this was not an argument about attempting to overcome a single barrier multiple times, but rather about the need to overcome multiple barriers to reach a habitat.

Because this is an important argument, I feel it would be better to make this point of view the purpose of the paper and make it a bit clearer.

**Do you want your identity to be public for this peer review?** For information about this choice, including consent withdrawal, please see our Privacy Policy

Reviewer #1: **Yes: ** Md. Fakhrul Islam

Reviewer #2: **Yes: ** Syed Shafat Hussain

Reviewer #3: No

Reviewer #4: No

---

## [Author Response · Author response to Decision Letter 1]

4 Jun 2025

Manuscript ID: PONE-D-24-53465

Dear Editor Dr Mizanur Rahman,

Thank you for the opportunity to revise our manuscript “Teaching fish new tricks: Repeated exposure to a velocity barrier improves passage performance ” (Manuscript ID PONE-D-24-53465) for consideration for publication in PLOS ONE. We appreciate the constructive comments provided by the two peer reviewers. We hope that our revision has suitably addressed all identified concerns.

Please see below for a point-by-point summary of changes as raised by reviewers (in RED). We have also provided both a tracked change and clean copy of the revised manuscript for your consideration. Any references to line numbers in our response to the reviewers are based on the tracked change version.

On behalf of the full authorship,

Yours sincerely,

Rachel Crawford

National Institute of Water and Atmospheric Research, Hamilton, New Zealand

rachelmbcrawford@gmail.com

Reviewer 1:

Thank you for the opportunity to review this manuscript. The paper is well-written and structured, and it addresses an important topic: how repeated exposure to a high-velocity water barrier affects the ability of juvenile Galaxias maculatus to pass through it. The experiment involved subjecting individual fish to the same high-speed water conditions in a raceway over five consecutive days. They found that the proportion of fish successfully passing the barrier increased significantly from 40% on Day 1 to 63% on Day 5. Time-to-event analysis revealed that fish passed the barrier more quickly by Day 5 than Day 1 but did not significantly improve approach or entry rates into the raceway. The study suggests that cognition and spatial memory play a role in improving passage performance through velocity barriers, but other factors, such as attraction flows, are also important. However, there are a few points that could be strengthened:

Thank you for your feedback. We have made revisions in response to your comments as described below.

1. A total of 40 fish were used in the experiment. Although this sample size may be adequate, a larger sample size could provide more robust results and increase the statistical power of the study.

While a larger sample size would produce more robust results, we were limited logistically, in terms of seasonal availability and amount of time it takes to conduct trials, which limited the number of fish we could use in this study.

2. The study focuses on one species (Galaxias maculatus), which may not fully represent cognitive responses in other fish species.

This is correct that Galaxias maculatus may not fully represent cognitive responses in other fishes. We hope to use this as a foundation for future work to look at cognitive abilities across more species in the future. We have also added in some discussion around this limitation (Lines 413-415).

3. While the study discusses cognition and memory, it does not explicitly assess fish personality traits (e.g., boldness or exploratory behavior), which could influence passage success.

In hindsight, individual fish personality is something we wish we had assessed. We have included some theories on how fish personality may have influenced passage success in our experimental raceway rather than just cognitive ability alone (Lines 443-453).

4. While water temperature was monitored, other environmental factors, such as dissolved oxygen and pH levels, were not explicitly addressed. Fluctuations in these factors could potentially influence fish behavior and passage performance.

While other environmental variables were not monitored, such as dissolved oxygen and pH levels, we expected the variability in these variables to be relatively minimal. The flume was open to the air and had highly turbulent flow (Reynolds number = 220, 588). Any differences in dissolved oxygen would be minimal and strongly correlated with temperature. For pH, the fish were very small relative to the volume of the experimental setup, and no other factors were present that would influence pH, as the system used raw, dechlorinated water. See lines 143-148.

5. The experimental raceway does not fully replicate natural stream conditions, which may affect generalizability.

This limitation has been discussed in the Discussion section (Lines 407-413). However, the controlled environment of the laboratory flume offers several advantages, including the ability to control for confounding factors. This allows us to more accurately isolate learning effects from other influences, such as variation in group size, as well as ensuring consistent flow conditions and water speeds across all trials. In the first instance we need to have a controlled experimental setup which can then inform field tests in the future.

6. The study does not assess whether repeated exposure induces stress or fatigue, which could influence fish behavior.

This is a valid point—measuring blood chemistry as a proxy for stress would be a valuable addition in future experiments.

Therefore, based on the concerns outlined above, I recommend that this manuscript be accepted for publication after major revision.

It is not feasible to repeat these experiments with a greater number of individuals or with additional species at this stage. If the editor considers these additions essential for publication in this journal, we are prepared to withdraw the manuscript. While we acknowledge that including additional species or increasing the sample size would strengthen the study, we believe the current work offers a valuable contribution to the literature and provides a solid foundation for future research.

Reviewer 2:

The study addresses an important ecological issue: the impact of anthropogenic barriers on fish migration. The findings have implications for conservation and management of freshwater ecosystems, particularly for small-bodied, non-salmonid species. The focus on Galaxias maculatus, an obligate migratory species, adds ecological relevance, as this species is widely distributed in the Southern Hemisphere and faces challenges from human-altered environments. Following points need to be addressed for improvement in manuscript.

Thank you for your suggestions and recommendations for the improvement of our manuscript.

1. At line 109, Elaborate reason why authors kept fish in 6 ppt salinity. Does it match the source of procurement? Give reason.

The use of 6 ppt salinity in quarantine tanks is to help prevent disease and kill any potential disease the fish may bring with them from the river. This is explained at line 125.

2. Expand the sample size and replicate the experiment across multiple groups or populations to improve the robustness of the findings.

While a larger sample size would produce more robust results, we were limited logistically, in terms of seasonal availability and amount of time it takes to conduct trials, which limited the number of fish we could use in this study.

3. Consider reusing individual fish in follow-up experiments to assess individual variability in learning and performance.

In hindsight, this would be a good suggestion, however, it was not feasible given the seasonality and timing constraints of this experiment. As suggested to Reviewer 1, in the future we would like to assess individual fish personality and take that into account in our experiments.

4. Isn't it better to consider reusing individual fish in follow-up experiments to assess individual variability in learning and performance.

This appears to make the same point as item 3, and as noted it was unfortunately not feasible to conduct further trials in the current study.

5. Provide specific recommendations for conservation and management based on the findings, such as how to improve fishway design or mitigate the impacts of barriers on fish migration.

We have added some additional discussion around this topic, please see lines 421-430.

6. The experimental setup, while controlled, may not fully replicate real-world conditions. For example, the raceway's simplified structure and uniform flow conditions do not account for the sample size (40 fish) is relatively small, and the study does not mention whether the experiment was replicated across multiple groups or populations. Replication would strengthen the robustness of the findings.

Similar to Reviewer 1’s comments, while a larger sample size would produce more robust results, we were limited logistically, in terms of seasonal availability and amount of time it takes to conduct trials, which limited the number of fish we could use in this study. Additionally, while only juvenile Galaxias maculatus were used in this study, care should be taken not to over-extrapolate findings to other populations or other fish species (Lines 413-415). We have discussed limitations to this study in the discussion section, lines 407-415.

7. Consider conducting follow-up experiments in more complex environments or with different types of barriers to assess the generalizability of the findings. Incorporate variability in flow conditions or barrier design to better mimic real-world scenarios. Include direct measures of spatial learning, memory, and behavioral traits (e.g., boldness, exploration) to strengthen the interpretation of results.

As suggested to Reviewer 1, this limitation has been discussed in the Discussion section (Lines 443-453). However, the controlled environment of the laboratory flume offers several advantages, including the ability to control for confounding factors. This allows us to more accurately isolate learning effects from other influences, such as variation in group size, as well as ensuring consistent flow conditions and water speeds across all trials. In the first instance we need to have a controlled experimental setup which can then inform field tests in the future.

The manuscript presents a well-designed and executed study with important implications for understanding fish passage through anthropogenic barriers. However, the ecological realism, exploration of behavioral mechanisms, and statistical reporting could be improved. Addressing these weaknesses would strengthen the manuscript and enhance its contribution to the field of freshwater ecology and conservation.

In lieu of any specific statistical comments from this reviewer, we are using Reviewer 3’s comments and have made changes in line with their recommendations.

Reviewer 3:

The manuscript presents a technically sound scientific investigation, supported by a detailed and rigorous methodology. It includes all the essential elements of a well-structured study, such as the description of the required permits and approvals for fish collection, experimental trials, setup and experimental protocol, as well as statistical analysis.

The results are presented clearly and in alignment with the methodology, allowing the reader to assess whether the objectives were achieved. Additionally, the analyses are well-founded, providing a robust interpretation of the data while transparently acknowledging the study’s limitations. In particular, the article highlights the ability of fish to improve their passage success with repeated exposures but also recognizes that it remains uncertain whether this learning is transferable to new obstacles.

A valuable aspect of the manuscript is that, in addition to addressing its research question, it provides relevant findings for the management of fish passage structures. It highlights challenges in fish attraction and entry, as well as the potential effect of shyness in some individuals, suggesting that management strategies should focus not only on the physical structure of passageways but also on improving fish attraction.

Overall, this study significantly contributes to the understanding of fish passage efficiency and provides valuable information for both science and the management of aquatic ecosystems.

Thank you very much for your feedback and your suggestions.

Editorial Recommendations

Some structural adjustments are necessary in the manuscript:

a) In the Modelling Binary Success section, when mentioning p-values, the text should correctly direct the reader to Table 2.

Thank you for pointing this out, this has now been corrected.

b) In line 278, "Table 2" should be changed to "Table 4."

Thank you for pointing this out, this has now been corrected.

c) It is recommended to reorganize the order of the paragraphs to maintain the logical sequence of the results, placing Table 3 before Table 4.

Thank you for pointing this out, this has now been corrected.

2. I consider that the statistical analysis in the manuscript has been conducted appropriately and rigorously. The authors applied a solid approach using generalized binomial generalized linear mixed models (GLMM) to model binary success in approach, entry, and successful passage, incorporating random effects to capture individual variability in fish. Additionally, they employed Cox regression to assess event rates (approach rate, entry rate, and passage rate), allowing for the interpretation of the effects of repeated exposure over time and generating Hazard Ratios (HR) to measure the instantaneous probability of success at each threshold of the experimental channel.

However, I found an inconsistency in the application of the Hosmer-Lemeshow goodness-of-fit test. In the methodology, it is stated that this test was used to evaluate model fit in the GLMM for all three events (approach, entry, and successful passage), but in the results, its use is only reported for the entry model. This raises questions about whether it was also applied to the other two events or if its reporting was omitted.

Furthermore, I consider that the Hosmer-Lemeshow test is not the most appropriate for evaluating the fit of GLMM models, as it does not account for the random effects characteristic of these models. I suggest using more suitable methods, such as residual inspection, AIC/BIC comparison, or the simulation of predicted versus observed values. If a new model fit method is adopted, I recommend applying it uniformly across all events and including the corresponding results in the results section to ensure consistency with what is described in the methodology.

Despite this observation, I believe that the analyses used in the study are appropriate for addressing the research question and allow for well-supported conclusions.

Thank you for pointing this out. After doing more research, we found that you are correct and the AIC comparison is more appropriate for GLMM models with random effects. Using the AIC comparison, the GLMM model chosen for each threshold meets the AIC selection criteria (Lines 247-251).

3. In the manuscript, the authors indicate that the data will be available through the Zenodo digital repository and provide the corresponding doi: 10.5281/zenodo.14194951. However, I was unable to access this DOI or find an active link.

I suggest that the authors verify the availability of the link and confirm that the data are correctly deposited in the repository to ensure open access and transparency in the publication.

The data are now available via the Zenodo digital repository: https://doi.org/10.5281/zenodo.14194951

4. The manuscript is written clearly, without ambiguities, and in a comprehensible level of English. During my review, I did not identify any typographical or grammatical errors that would affect the understanding of the content. However, since English is not my native language, my assessment of full compliance with linguistic standards is limited.

Reviewer 4:

Were the juveniles caught before they had completed their upstream migration to their habitat? Or was the capture location their habitat?

This may be considered one of the reasons why "fish failed to enter the raceway each day."

Juvenile fish were captured at the beginning of their upstream migration to control for migratory urge across individuals (Lines 120-122).

L117

Did they reuse the waterway described in (39), or did they use the features of (39)?

I would like you to add a little more explanation about the introduction of the waterway so that it can be understood without reading (39).

We have clarified in text that this study used the same flume and same setup as described in (39).

If PVC is an abbreviation for something

---

## [Decision Letter · Decision Letter 1]

22 Jun 2025

Dear Dr.Crawford,

Thank you for submitting your manuscript to PLOS ONE. After careful consideration, we feel that it has merit but does not fully meet PLOS ONE’s publication criteria as it currently stands. Therefore, we invite you to submit a revised version of the manuscript that addresses the points raised during the review process.

We look forward to receiving your revised manuscript.

Kind regards,

Mizanur Rahman, Ph.D.

Academic Editor

PLOS ONE

Journal Requirements:

Reviewers' comments:

Reviewer's Responses to Questions

**Comments to the Author**

Reviewer #1: All comments have been addressed

Reviewer #2: All comments have been addressed

Reviewer #3: (No Response)

2. Is the manuscript technically sound, and do the data support the conclusions?

Reviewer #1: Yes

Reviewer #2: Yes

Reviewer #3: Partly

3. Has the statistical analysis been performed appropriately and rigorously?

Reviewer #1: Yes

Reviewer #2: Yes

Reviewer #3: I Don't Know

4. Have the authors made all data underlying the findings in their manuscript fully available?

Reviewer #1: Yes

Reviewer #2: Yes

Reviewer #3: Yes

5. Is the manuscript presented in an intelligible fashion and written in standard English?

Reviewer #1: Yes

Reviewer #2: Yes

Reviewer #3: Yes

Reviewer #1: I appreciate the efforts made by the authors to provide a more concise version that better highlights the important findings. However, the author addresses the comments very carefully, and I gladly recommend that this manuscript be accepted for publication.

Reviewer #2: (No Response)

Reviewer #3: 1. Although I initially assumed that the authors had implemented the requested corrections regarding model fitting, as the methods section explicitly mentions the use of the Akaike Information Criterion (AIC) to select the most appropriate models for the three thresholds (approach, entry, and passage success), upon reviewing the results section, it is unclear whether the models presented were actually selected based on this criterion. Instead, the Hosmer-Lemeshow test is mentioned, despite not being previously referenced, which introduces a methodological inconsistency. This lack of alignment between the methods and results sections prevents verification that the reported models truly represent the best fit according to AIC. Clarification of this issue in the results section is recommended.

That said, the authors have adequately addressed all editorial recommendations. Table and figure references are now accurate and consistent with the narrative. The paragraph order has been corrected for logical flow, and all statistical results are clearly linked to the corresponding tables. No inconsistencies remain in that regard.

2. I marked "partially" because, although the study presents a clear structure and a solid experimental design, there remains an unresolved methodological issue that limits a more positive evaluation. As previously noted to the authors, the application of the model selection criterion (AIC) has not been adequately clarified, leading to a lack of coherence between the methodology and the results sections. This omission prevents confirmation that the reported statistical models indeed reflect the best-fitting ones. While the results are well-articulated and the study provides valuable insights for fish passage management, the authors must address this methodological inconsistency in order for the manuscript to be considered technically complete.

3. I select “Don’t know” because, although the statistical approach is appropriate for the research question, there remains a methodological inconsistency between the model selection criterion described in the methods section (AIC) and the one reported in the results (Hosmer-Lemeshow test). This discrepancy prevents a clear confirmation that the statistical analysis was applied rigorously and consistently.

4. In this revised version of the manuscript, the link provided to access the data through the Zenodo digital repository (DOI: https://doi.org/10.5281/zenodo.14194951) is active and allows full download of the dataset. This correction ensures open access and enhances the transparency and reproducibility of the results presented in the study.

I would like to encourage the authors to make the necessary adjustments so that their study can be shared with the scientific community and contribute to guiding and managing fish passage in rivers impacted by dams and reservoirs.

**Do you want your identity to be public for this peer review?** For information about this choice, including consent withdrawal, please see our Privacy Policy

Reviewer #1: No

Reviewer #2: **Yes: ** Syed Shafat Hussain

Reviewer #3: No

---

## [Author Response · Author response to Decision Letter 2]

26 Jun 2025

Manuscript ID: PONE-D-24-53465

Dear Editor Dr Mizanur Rahman,

Thank you for the opportunity to revise our manuscript “Teaching fish new tricks: Repeated exposure to a velocity barrier improves passage performance ” (Manuscript ID PONE-D-24-53465) for consideration for publication in PLOS ONE. We appreciate the constructive comments provided by the two peer reviewers. We hope that our revision has suitably addressed all identified concerns.

Please see below for a point-by-point summary of changes as raised by reviewers (in RED). We have also provided both a tracked change and clean copy of the revised manuscript for your consideration. Any references to line numbers in our response to the reviewers are based on the tracked change version.

On behalf of the full authorship,

Yours sincerely,

Rachel Crawford

National Institute of Water and Atmospheric Research, Hamilton, New Zealand

rachelmbcrawford@gmail.com

Response to Reviewers:

Reviewer 1:

I appreciate the efforts made by the authors to provide a more concise version that better highlights the important findings. However, the author addresses the comments very carefully, and I gladly recommend that this manuscript be accepted for publication.

Thank you very much for your feedback throughout this process, we appreciate the time that was put into it.

Reviewer 2: (No Response)

Reviewer 3:

1. Although I initially assumed that the authors had implemented the requested corrections regarding model fitting, as the methods section explicitly mentions the use of the Akaike Information Criterion (AIC) to select the most appropriate models for the three thresholds (approach, entry, and passage success), upon reviewing the results section, it is unclear whether the models presented were actually selected based on this criterion. Instead, the Hosmer-Lemeshow test is mentioned, despite not being previously referenced, which introduces a methodological inconsistency. This lack of alignment between the methods and results sections prevents verification that the reported models truly represent the best fit according to AIC. Clarification of this issue in the results section is recommended.

That said, the authors have adequately addressed all editorial recommendations. Table and figure references are now accurate and consistent with the narrative. The paragraph order has been corrected for logical flow, and all statistical results are clearly linked to the corresponding tables. No inconsistencies remain in that regard.

We apologize for the lack of clarity in the manuscript regarding our switch to using AIC for model selection. In the revised version, we did adopt AIC as our primary model selection criterion; however, we overlooked removing previous references to the Hosmer-Lemeshow test. We have now corrected this by removing all mentions of the Hosmer-Lemeshow test and have added a table (lines 247–257) outlining our model selection process along with the associated AIC scores for transparency.

2. I marked "partially" because, although the study presents a clear structure and a solid experimental design, there remains an unresolved methodological issue that limits a more positive evaluation. As previously noted to the authors, the application of the model selection criterion (AIC) has not been adequately clarified, leading to a lack of coherence between the methodology and the results sections. This omission prevents confirmation that the reported statistical models indeed reflect the best-fitting ones. While the results are well-articulated and the study provides valuable insights for fish passage management, the authors must address this methodological inconsistency in order for the manuscript to be considered technically complete.

We apologize for the inconsistency, and hope that our changes to the manuscript have made this clearer. Please see lines 247-257 for a description of our AIC model selection process, along with a table summarizing the AIC scores.

3. I select “Don’t know” because, although the statistical approach is appropriate for the research question, there remains a methodological inconsistency between the model selection criterion described in the methods section (AIC) and the one reported in the results (Hosmer-Lemeshow test). This discrepancy prevents a clear confirmation that the statistical analysis was applied rigorously and consistently.

We apologize for the inconsistency, and hope that our changes to the manuscript have made this clearer. Please see lines 247-257.

4. In this revised version of the manuscript, the link provided to access the data through the Zenodo digital repository (DOI: https://doi.org/10.5281/zenodo.14194951) is active and allows full download of the dataset. This correction ensures open access and enhances the transparency and reproducibility of the results presented in the study.

Thank you very much for your feedback throughout this process, we greatly appreciate the time that was put into it.

---

## [Decision Letter · Decision Letter 2]

16 Jul 2025

Teaching fish new tricks: Repeated exposure to a velocity barrier improves passage performance

PONE-D-24-53465R2

Dear Dr. Crawford,

We’re pleased to inform you that your manuscript has been judged scientifically suitable for publication and will be formally accepted for publication once it meets all outstanding technical requirements.

Kind regards,

Mizanur Rahman, Ph.D.

Academic Editor

PLOS ONE

Additional Editor Comments (optional):

Reviewers' comments:

Reviewer's Responses to Questions

**Comments to the Author**

Reviewer #3: All comments have been addressed

2. Is the manuscript technically sound, and do the data support the conclusions?

Reviewer #3: Yes

3. Has the statistical analysis been performed appropriately and rigorously?

Reviewer #3: Yes

4. Have the authors made all data underlying the findings in their manuscript fully available?

Reviewer #3: Yes

5. Is the manuscript presented in an intelligible fashion and written in standard English?

Reviewer #3: Yes

Reviewer #3: The authors have adequately addressed a key concern raised in the previous round regarding the methodological coherence between the description of model selection using AIC and the presentation of results. In the revised version, they clearly indicate that model selection for binary success outcomes was based on the corrected Akaike Information Criterion (AICc), and this is now consistent across the Methods, Results (Tables 1 and 3), and Discussion sections. The selected models are appropriate for the research questions posed, and their use of GLMMs and Cox mixed-effects models aligns well with the binary and time-to-event data structure.

The statistical analyses have been performed rigorously, with thoughtful inclusion of random effects to account for individual variability, and proper handling of censored data. The consistency between the methodological framework and the statistical outputs strengthens the reliability of the conclusions.

Additionally, the conclusions are appropriately drawn from the data and analyses. The authors maintain a balanced interpretation, acknowledging both the statistically significant effects and the limitations of the observed trends, particularly in the context of real-world fish passage structures. Their findings contribute valuable insights into how cognitive and physical traits may interact with hydraulic barriers, which is highly relevant for future design and assessment of fishways.

Therefore, I can now state that the manuscript is technically sound and methodologically coherent, and I commend the authors for the thoughtful revisions that enhance the clarity and scientific robustness of the work.

**Do you want your identity to be public for this peer review?** For information about this choice, including consent withdrawal, please see our Privacy Policy

Reviewer #3: No

---

## [Editor Report · Acceptance letter]

PONE-D-24-53465R2

PLOS ONE

Dear Dr. Crawford,

I'm pleased to inform you that your manuscript has been deemed suitable for publication in PLOS ONE. Congratulations! Your manuscript is now being handed over to our production team.

Kind regards,

on behalf of

Dr. Mizanur Rahman

Academic Editor

PLOS ONE